# Why Should the Server Do It All?: A Scalable, Versatile, and Model-Agnostic Framework for Server-Light DNN Inference over Massively Distributed Clients via Training-Free Intermediate Feature Compression

## Abstract

Modern DNNs often rely on edge–cloud model partitioning (MP), but widely used schemes fix shallow, static split points that underutilize edge compute and concentrate latency and energy on the server. The problem is exacerbated in autoregressive (AR) LLM inference, where per-token forward passes repeatedly generate bulky intermediate features (IFs). We introduce SLICER, a retraining-free, architecture-agnostic framework that compresses IFs to reduce both communication and server load in split computing. SLICER combines (i) asymmetric top-K filtering (ATKF) to sparsify low-magnitude activations, (ii) magnitude-splitting (MS) to group the remaining non-zeros into equal-cardinality blocks, and (iii) adaptive bit quantization (ABQ) that selects per-block bitwidths under a distortion budget. Across standard vision and LLM workloads (e.g., ImageNet/COCO; HellaSwag, PIQA, ARC-E/C, GSM8K, HumanEval), SLICER reduces uplink volume by up to 10× and server GPU time by up to 4.4×, while keeping task quality within 0–3 pp of baseline. In multi-device settings and AR LLMs, SLICER scales by shifting meaningful compute to the edge and lowering bits-per-token and server time per token, stabilizing per-step traffic. The codec attaches to off-the-shelf models without retraining or architectural changes, offering a plug-and-play path to scalable, low-latency distributed inference. Code is provided in the supplementary material.

## 1 Introduction

Deep neural network (DNN) has advanced at an unprecedented pace, primarily enabled by the ability to train deeper neural networks. This breakthrough was largely facilitated by *residual learning* He et al. (2016), where skip connections mitigate the vanishing gradient problem and allow the convergence of networks with hundreds or even thousands of layers. Building on this foundation, both the depth and width of neural networks have been scaled, achieving state-of-the-art (SOTA) performance Krizhevsky et al. (2012); Touvron et al. (2021); Vaswani et al. (2017); Gu et al. (2021); Gu & Dao (2023); Brown et al. (2020); Hoffmann et al. (2022). However, deploying these high-performance models across diverse hardware platforms remains a significant challenge, as their computational and memory demands often surpass the capabilities of target devices.

### 1.1 Related Works and Motivations

**DNN Model Partitioning (MP).** Within edge computing (EC), a common baseline is to transmit raw inputs to a back-end server that executes the entire DNN Matsubara et al. (2022b). While this preserves model accuracy and offloads device compute, it can cause (1) server congestion as the number of clients grows Kang et al. (2017), (2) higher end-to-end (E2E) latency under adverse networks Bakhtiarnia et al. (2023), and (3) underutilization of increasingly capable edge accelerators Matsubara et al. (2022a). Consequently, efficient distributed inference techniques such as model partitioning (MP) are essential to bridge the resource gap for deploying neural models on edge

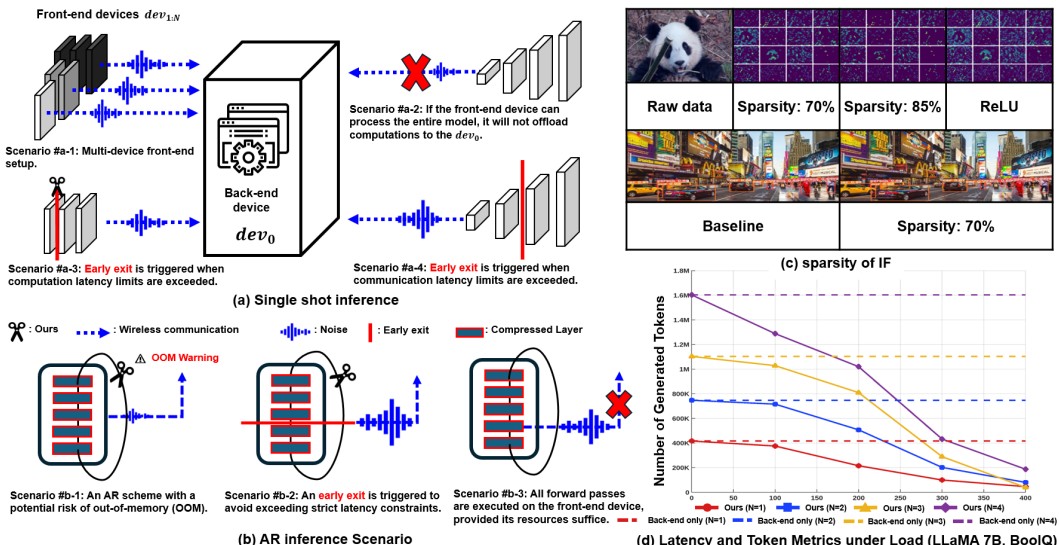

Figure 1: (a) Single-shot inference model where multiple front-end devices offload tasks to a shared back-end based on latency constraints. (b) AR inference scenarios illustrating early exits triggered by memory or latency limits. (c) Visualization of sparsity in IFs, a key technique for compression. (d) Performance metrics demonstrating that our approach reduces the back-end server's load as the number of front-end devices increases.

devices. To mitigate these issues, *split computing* (SC)—a widely studied *subclass* of MP—partitions the network into: (i) early layers executed on the front-end (edge) and (ii) the remaining layers on the back-end (server). Instead of raw inputs, the front-end transmits *intermediate features* (IFs), which can reduce bandwidth and E2E latency. With an appropriate split point, SC may also reduce the exposure of raw content in transit Matsubara et al. (2022b).

**Cloud-Centric Bias in MP and Its Amplification in LLMs.** A critical flaw in many existing MP policies is their reactive, cloud-centric nature. In response to deteriorating wireless links, they often default to shifting the split point closer to the input Bakhtiarnia et al. (2023); Kim & Ko (2023), a strategy that offloads the vast majority of the computational burden to the server. While this may reduce communication latency for a single client, it becomes an unscalable, server-heavy approach in realistic multi-device deployments, amplifying queueing delays and operational costs Matsubara et al. (2022a); Pagliari et al. (2020).

This problem is severely magnified in modern LLM services. Here, naive, server-centric offloading underutilizes powerful edge devices and quickly overloads shared servers Patel et al. (2024); Sung et al. (2025), an effect we quantify in Fig. 1(d). The challenge is further exacerbated by the nature of autoregressive (AR) inference, which introduces two unique problems: (1) per-token full-stack execution diminishes the amortized benefits of shallow splits, and (2) accumulating traffic and state mean the total data volume grows with the output length.

Unfortunately, prior MP work on IF compression is almost entirely vision-centric, focusing on single-shot payload reduction Ahuja et al. (2023); Furtuanpey et al. (2024); Matsubara et al. (2022c); Duan & Zhu (2022); Datta et al. (2022). These methods are ill-suited for the token-by-token nature of AR workloads, where continuous, per-step bandwidth savings are crucial for maintaining low latency. The limitations of existing cloud-centric policies and the unsuitability of vision-focused codecs for AR workloads highlight a clear need for a new approach.

## 1.2 CONTRIBUTIONS

To address this gap, in this work, we introduce *SLICER*, a training-free, task-agnostic framework for split computing that couples a lightweight IF codec with a predictive, constraint-aware configuration. Our key technical contributions and breakthroughs in this work include the followings:

- **Lightweight IF Compression & Balanced Utilization.** We propose a lightweight IF compression scheme that effectively exploits sparsity, suppresses quantization error, and optimizes bit-width. Implemented with fused CUDA kernels for low overhead, its compression ratio is precisely tunable to match diverse model partitioning (MP) scenarios.

- **Generality Across Vision & NLP Tasks.** Our framework is designed to be task-agnostic. A single codec handles both vision and NLP workloads, which eliminates redundant, specialized components and simplifies deployment across heterogeneous devices.

- **Plug-and-Play Integration.** Our codec is plug-and-play, attaching to off-the-shelf models without retraining or architectural changes. Validated across extensive vision and LLM benchmarks, it reduces server-side latency by up to 4.4× and shrinks bandwidth substantially while preserving model accuracy—all without requiring model modification.

## 2    OUR APPROACH: SLICER

### 2.1    PARTITION LAYER SELECTION STRATEGY FOR SERVER-LIGHT DNN INFERENCE

**DNN Inference Latency.** Consider a DNN $\mathbb{L} = \{\mathbb{L}_1, \mathbb{L}_2, \ldots, \mathbb{L}_{\hat{\ell}}\}$ consisting of $\hat{\ell}$ layers, where $\mathbb{L}_\ell$ denotes the $\ell$-th layer. Define the partial network $\mathbb{L}_{1:\ell}$ as the sequential composition of layers $\mathbb{L}_1, \ldots, \mathbb{L}_\ell$. The IF is given by

$$\mathbf{x}_\ell = \mathbb{L}_{1:\ell}(\mathbf{x}).$$

Given a selected split point $\ell$, the IF $\mathbf{x}_\ell$ is transmitted to the back-end device ($\text{dev}_0$). Each front-end device communicates over a noisy wireless channel whose quality may vary from node to node. Thus, we define the communication latency using the $\varepsilon$-outage latency Yun et al. (2022):

$$L_c(\ell, R) = \frac{B(\mathbf{x}_\ell)}{R} \left\lceil \frac{\ln \varepsilon}{\ln P_o(R)} \right\rceil + \zeta(\mathbf{x}_\ell), \qquad P_o(R) = 1 - \exp\left(-\frac{2^{R/W}-1}{\gamma}\right). \tag{1}$$

where $B(\mathbf{x}_\ell)$ is the bit-length of the IF, $R$ denotes the transmission rate, $W$ the channel bandwidth, and $\gamma$ the average signal-to-noise ratio (SNR). Additionally, $\zeta(\mathbf{x}_\ell)$ represents the time required to encode the IF $\mathbf{x}_\ell$ prior to transmission. We define the computation latency $L_d(\mathbb{L}_{1:\ell})$ as the processing time incurred by the front-end device[1]. Hence, the total latency is given by:

$$L(\ell, R) = L_d(\mathbb{L}_{1:\ell}) + L_c(\ell, R). \tag{2}$$

**DNN Partitioning Approach with Multiple Devices.** We consider a deployment consisting of one back-end device and $N$ front-end devices $\{\text{dev}_0; \text{dev}_1, \ldots, \text{dev}_N\}$.

- **Back-end device ($\text{dev}_0$):** Executes all layers beyond the split point and completes the inference.

- **Front-end devices ($\text{dev}_{1:N}$):** Each front-end device processes layers up to the chosen split point and transmits the IF to $\text{dev}_0$ for further processing.

**Single-Shot DNN Inference Scenario.** *Scenario #a-1* in Fig. 1(a) illustrates the multi-device scenario. A front-end device, $\text{dev}_i$, with memory budget $\mathcal{M}$, aims to execute the maximum number of layers while satisfying the latency constraint $\mathcal{D}$. For a DNN of $\hat{\ell}$ layers, let $m(\mathbb{L}_{1:\ell})$ denote the cumulative memory requirement of layers $1:\ell$. We select the deepest feasible split point $\ell^*$ that satisfies both latency and memory constraints:

$$\ell^* = \max\left\{\ell \in \{1, \ldots, \hat{\ell}\} \ \middle| \ m(\mathbb{L}_{1:\ell}) \leq \mathcal{M}, L(\ell, R) \leq \mathcal{D}\right\}. \tag{3}$$

The front-end device executes layers $\mathbb{L}_{1:\ell^*}$ and transmits the IF to $\text{dev}_0$, which completes inference by executing layers $\mathbb{L}_{\ell^*+1:\hat{\ell}}$. If the wireless link degrades such that $L(\ell^*, R) > \mathcal{D}$, Eq. 3 automatically selects a shallower split point, thereby satisfying both latency and memory constraints.

---

[1]The back-end device bypasses complex delay modeling by assigning each front-end a delay constraint that incorporates the back-end's own status.

**Autoregressive DNN Inference Scenario.** AR inference iteratively feeds the model's outputs back into its inputs, requiring repeated forward passes. Under these conditions, the partition strategy described in Eq. 3 may become impractical as memory consumption increases with each decoding step. To manage this memory growth, we adopt a compressed network $\lessdot_{1:\hat{\ell}}$. Let $C(\mathbb{L}_{1:\hat{\ell}}; P)$ denote the overall performance after applying the compression parameter $P$ to all layers. Accounting for the extra memory required for repeated forward passes, $\mathcal{M}_{ar}$, we aim to find the parameter $P_{\max}$ that maximizes performance:

$$P_{\max} = \arg\max_{P} C(\mathbb{L}_{1:\hat{\ell}}; P) \quad \text{s.t.} \quad m(\lessdot_{1:\hat{\ell}}) + \mathcal{M}_{ar} \leq \mathcal{M}. \tag{4}$$

Satisfying this constraint keeps the compressed network memory-feasible throughout AR decoding, preventing out-of-memory (OOM) errors while maintaining high performance.

For the first $(w-1)$ steps, the $\text{dev}_i$ processes all compressed layers locally (i.e., $\lessdot_{1:\hat{\ell}}$). This incurs a per-step cost of $L_d(\lessdot_{1:\hat{\ell}})$ on $\text{dev}_i$. On the final ($w$-th) step, however, the $\text{dev}_i$ executes only up to $\ell$, then offloads the IF $\mathbf{x}_\ell^w$ to $dev_0$. Consequently, the total latency is

$$L_{\text{AR}}(\ell, w, R) = \underbrace{(w-1)\, L_d(\lessdot_{1:\hat{\ell}})}_{\text{full passes}} + \underbrace{L_d(\lessdot_{1:\ell})}_{\text{partial pass}} + \underbrace{L_c(\ell, R)}_{\text{offloading IF}}. \tag{5}$$

Eq. 5 represents the total latency for a split point $(\ell, w)$. By requiring this latency to remain below the budget $\mathcal{D}$ and the offloaded feature size to stay within the per-step memory cap $\mathcal{M}_{ar}$, we then select the deepest admissible split point $(w^*, \ell^*)$ as follows:

$$(w^*, \ell^*) = \max_{w, \ell}\Big\{ (w, \ell) \ \Big| \ L_{\text{AR}}(\ell, w, R) \leq \mathcal{D}, \ B(\mathbf{x}_\ell^w) \leq \mathcal{M}_{ar} \Big\}, \tag{6}$$

which consistently meets both the latency and the memory constraints while maximizing computation on the front-end devices, $dev_{i:N}$. The strategy in Eq. 6 works as follows. When the memory or latency budget becomes tighter, each $dev_i$ moves to the next shallower split point *Scenario #b-1, #b-2* in Fig. 1(a), maximizing computation on the device. In contrast, when resources are sufficient, there is no need to offload to $dev_0$ *Scenario #b-3*, reducing round-trip traffic and avoiding unnecessary queueing delays. In this way, the system adapts on the fly to channel quality and device heterogeneity, balancing edge computation with minimal server assistance throughout AR inference.

# 3 FURTHER BREAKTHROUGH: INTERMEDIATE FEATURE COMPRESSION IN SLICER

Shrinking the IF is essential for practical model partitioning, especially in today's dynamic environments running mixed vision and NLP workloads. An effective compression scheme for this context must meet four strict criteria that conventional methods fail to satisfy simultaneously: (1) guaranteeing an exact compression ratio, (2) controlling this ratio at runtime, (3) operating at real-time speed, and (4) remaining task-agnostic.

Existing techniques fall short, as they were originally developed for a different purpose: static model weight compression for inference optimization, not dynamic IF compression. This fundamental design mismatch makes them unable to accurately capture a target compression ratio, a strict requirement for dynamic MP but a loose one for offline weight pruning. Moreover, since they are designed for a one-time, offline application, they inherently lack the mechanisms to dynamically control the compression level at runtime. Finally, many of these approaches are tailored to specific data modalities (e.g. vision), failing the task-agnostic requirement.

To meet all four demands, we propose a novel, unified, and training-free pipeline. Our asymmetric top-K filtering (ATKF) provides deterministic control for an exact compression target, while our adaptive bit quantization (ABQ) uses lightweight integer operations for real-time, per-block bit selection. Crucially, our entire task-agnostic framework, combined with magnitude-splitting (MS), delivers the precise, fast, and universally applicable control required for modern MP workloads. A comprehensive comparison highlighting these distinctions is provided in Appendix D. In addition, a detailed example of each stage can be found in the appendix A.

### 3.1 ASYMMETRIC TOP-K FILTERING (ATKF)

Let $x \in \mathbb{R}^{H \times W}$ with $N = HW$ elements and target sparsity $s \in [0, 1]$ and asymmetry factor $\lambda \in [0, 1)$. Define

$$k_{\text{keep}} = \lfloor (1 - s)\, N \rfloor, \qquad \tau = k_{\text{keep}}\text{-th largest } |x_{i,j}|, \qquad \tau_{+} = (1 + \lambda)\, \tau, \qquad \tau_{-} = -(1 - \lambda)\, \tau.$$

Strictly retained indices are

$$S_{\text{strict}} = \{(i, j) \mid x_{i,j} > \tau_{+} \text{ or } x_{i,j} < \tau_{-} \}.$$

If $|S_{\text{strict}}| < k_{\text{keep}}$, randomly select additional indices $T$ from $\{(i, j) \mid |x_{i,j}| = \tau\}$ until the total number of retained elements equals $k_{\text{keep}}$, i.e., $|S_{\text{strict}} \cup T| = k_{\text{keep}}$.

$$\mathcal{F}_{i,j}(x, s, \lambda) = \begin{cases} x_{i,j}, & (i, j) \in S_{\text{strict}} \cup T, \\ 0, & \text{otherwise.} \end{cases} \tag{7}$$

Eq. 7 keeps the $k_{\text{keep}}$ largest-magnitude coefficients and sets the remaining entries to zero, thereby achieving the exact nonzero count $k_{\text{keep}}$ (i.e., a nonzero fraction $1 - s$) while preserving shape. Note that if ties at $|x| = \tau$ are insufficient, we continue filling from the next lower magnitudes in descending $|x|$ (breaking ties at random) until $|S_{\text{strict}} \cup T| = k_{\text{keep}}$. The asymmetry factor $\lambda$ shifts the positive and negative cut-offs. Because only low-magnitude coefficients are suppressed—and ties at $|x| = \tau$ are resolved by random sampling—the split filter achieves the target sparsity with minimal impact on model accuracy and communication cost.

### 3.2 MAGNITUDE-SPLITTING (MS) WITH EQUAL CARDINALITY

To further reduce communication costs while preserving structural information, we apply a magnitude-based partitioning of the non-zero entries. Unlike typical thresholding schemes that define uniform intervals in the value domain, our approach ensures that each sub-tensor receives *same number of non-zero entries* in sorted order.

**Pre-Step: Sign Separation.** Given an IF $X \in \mathbb{R}^{N \times K}$ after *ATKF*, define

$$X^{(+)} = \max(X, 0), \quad X^{(-)} = \max(-X, 0), \quad X = X^{(+)} - X^{(-)}.$$

The remainder of this section is executed twice—once for $X^{(+)}$ and once for $X^{(-)}$-with identical logic.

**Step 1. Flatten, Collect non-zeros, Sort by magnitude.** Let $X^{(s)}$ be either sign ($s \in \{+, -\}$) and $X_{\text{flat}}^{(s)} \in \mathbb{R}^{T}$ its flattened view ($T = NK$)[2]. Non-zero indices $\Omega^{(s)} = \{\, i \mid X_{\text{flat}}^{(s)}(i) \neq 0 \}$ are extracted and sorted in descending absolute value:

$$(X_{\text{nz}}^{(s)}, \boldsymbol{\pi}^{(s)}) = \text{Sort}\big(|X_{\text{flat}}^{(s)}(i)| \mid i \in \Omega^{(s)}\big) \ \ (\textit{sorted descendingly}).$$

**Step 2. Equal-Cardinality partitioning.** For a target block count $M^{(s)}$, set $\mathcal{K}^{(s)} = \lfloor |\Omega^{(s)}|/M^{(s)} \rfloor$. Then, we can define the index ranges $I_m^{(s)}$ for $m = 1, \ldots, M^{(s)}$. Each block is re-inserted into a zero-initialized tensor $\mathbf{y}$ at positions indicated by $\boldsymbol{\pi}^{(s)}(i)$.

**Step 3. Compressed Sparse Row (CSR) Encoding.** Every block $X_m^{(s)}$ is stored using the CSR format: $\text{CSR}(X_m^{(s)}) = (\mathbf{v}_m^{(s)}, \mathbf{c}_m^{(s)}, \mathbf{r}_m^{(s)})$.

---

[2]For a general IF $X \in \mathbb{R}^{N \times K}$, ATKF is applied to the flattened tensor of length $T = NK$; thus $k_{\text{keep}} = \lfloor (1 - s)T \rfloor$.

### 3.2.1 Adaptive Bit Quantization (ABQ)

**Asymmetric Integer Quantization (AIQ).**  After MS, the non-zero values $\mathbf{v}_m$ in each block are quantized independently using Asymmetric Integer Quantization (AIQ) with a computed zero-point and $q_m$ bits:

$$\widehat{\mathbf{v}}_m = \left\lceil \frac{\mathbf{v}_m}{o_m} + \mathbf{z}_m \right\rceil, \qquad o_m = \frac{v_m^{\max} - v_m^{\min}}{2^{q_m} - 1}, \qquad \mathbf{z}_m = \left\lceil \frac{v_m^{\min}}{o_m} \right\rceil,$$

where $v_m^{\max} = \max(\mathbf{v}_m)$ and $v_m^{\min} = \min(\mathbf{v}_m)$. Since MS has already pruned extreme outliers, this per-block AIQ step introduces only minor distortion. The server reconstructs the tensor $\tilde{X}$ by reversing this process:

$$\tilde{X} = \Sigma \, \mathrm{csr}^{-1}(\tilde{\mathbf{v}}_m, \mathbf{c}_m, \mathbf{r}_m), \quad \tilde{\mathbf{v}}_m = (\hat{\mathbf{v}}_m - \mathbf{z}_m) \cdot o_m. \tag{8}$$

**ABQ as a Rate-Distortion Solver.**  To determine the optimal bit-width $q_m$ for each block, ABQ uses a lightweight "integer mismatch" metric, $DS(\cdot)$, instead of MSE (see Appendix B):

$$DS(T_1, q_1, T_2, q_2) = \frac{1}{n} \sum \left| \left\lfloor \frac{T_1}{2^{(q_1 - q_2)}} \right\rfloor - T_2 \right|. \tag{9}$$

The ABQ algorithm greedily lowers the bit precision $q$ from an initial maximum $q_{\mathrm{bit}}$ until a user-defined distortion threshold $\delta$ is met. Formally, we solve:

$$\min_{q \le q_{\mathrm{bit}}} q \quad \text{s.t.} \quad DS(\widehat{\mathbf{T}}_0, q_{\mathrm{bit}}, \widehat{\mathbf{T}}, q) \le \delta,$$

where $\widehat{\mathbf{T}}_0$ is the reference integer tensor. Because this search operates entirely on integers, it is computationally lightweight, allowing ABQ to efficiently balance bit usage and distortion at runtime.

## 3.3 Constraint-Aware Predictive Configuration (SLICER-Search)

Given real-time budgets $(\mathcal{D}, \mathcal{M}, \mathcal{M}_{\mathrm{ar}}, \mathcal{M}_{\mathrm{buf}}, R)$ and a candidate split (single-shot: $\ell$; AR: $(\ell, w)$), we choose compression knobs $\theta = \{s, M^{(+)}, M^{(-)}, \lambda\}$ to (i) minimize accuracy loss while (ii) strictly satisfying latency and memory/throughput constraints. ABQ per-block bitwidths $\mathbf{q}^{(s)} = \{q_m^{(s)}\}$ are determined by an integer-only distortion guard $\delta$ (Sec. 3.2.1).

**Predictive Bit/Time Model.**  Let the IF after ATKF/MS have $T = NK$ elements in total. ATKF fixes the nonzero count as $k_{\mathrm{keep}} = \lfloor (1 - s)T \rfloor$. MS then partitions nonzeros of each sign into $M^{(s)}$ equal-cardinality blocks, so each block has a known cardinality $|\Omega_m^{(s)}|$. Because ABQ selects $q_m^{(s)}$ adaptively from data, we predict the payload by a conservative upper bound that sets $q_m^{(s)} = q_{\mathrm{bit}}$:

$$\widehat{B}^{\mathrm{UB}}(\mathbf{x}_\ell; \theta) = \underbrace{\sum_{s \in \{+, -\}} \sum_{m=1}^{M^{(s)}} |\Omega_m^{(s)}| \, q_{\mathrm{bit}}}_{\text{value bits}} + \underbrace{B_{\mathrm{idx}}(N, K, M^{(+)}, M^{(-)}) + B_{\mathrm{meta}}(M^{(+)}, M^{(-)})}_{\text{CSR/header}}.$$

Accordingly, we redefine the communication latency from Eq. equation 1 as:

$$\widehat{L}_c(\ell, R; \theta, \delta) = \frac{\widehat{B}^{\mathrm{UB}}(\mathbf{x}_\ell; \theta)}{R} \left\lceil \frac{\ln \varepsilon}{\ln P_o(R)} \right\rceil + \widehat{\zeta}(\theta, \delta),$$

where $\widehat{\zeta}(\theta, \delta) = t_{\mathrm{ATKF}}(s, \lambda) + t_{\mathrm{MS}}(M^{(+)}, M^{(-)}) + \sum_{s,m} t_{\mathrm{ABQ}}(q_m^{(s)})$ is a lightweight integer-operation time model, estimated *pre-execution* from device-specific lookup tables (calibrated offline).

**Constraints.**  We adopt $\mathcal{M}_{\mathrm{ar}}$ as the per-step offload-buffer cap (same unit as bits; consistent with Eq. 6).

$$\text{single-shot:} \quad L(\ell, R; \theta, \delta) = L_d(\mathbb{L}_{1:\ell}) + \widehat{L}_c(\ell, R; \theta, \delta) \le \mathcal{D}, \quad m(\mathbb{L}_{1:\ell}) + m_{\mathrm{buf}}(\theta, \delta) \le \mathcal{M},$$

$$\text{AR:} \quad\quad\quad L_{\mathrm{AR}}(\ell, w, R; \theta, \delta) \le \mathcal{D}, \quad \widehat{B}^{\mathrm{UB}}(\mathbf{x}_\ell^w; \theta) \le \mathcal{M}_{\mathrm{ar}}.$$

**Hierarchical Search Policy**   To effectively navigate the complex configuration space under strict resource constraints, we employ a three-tier search strategy. This policy is governed by an "Accuracy-First" principle, designed to exhaust strictly non-destructive or low-distortion optimizations before resorting to aggressive workload shifts:

1. **Tier 1: Structural Optimization (Tune $M$).** The search initiates by adjusting the block counts $(M^{(+)}, M^{(-)})$ to optimize the trade-off between quantization granularity and metadata overhead. This step is prioritized as it enhances the coding efficiency of the bitstream structure—minimizing the header cost relative to the payload—without pruning any activation values, thereby preserving the original information content.

2. **Tier 2: Controlled Distortion (Increase $s$).** If the bandwidth budget remains unsatisfied after structural tuning, the policy incrementally increases the sparsity level $s$. This stage introduces *controlled distortion* by selectively zeroing out low-magnitude activations. By prioritizing the removal of less significant features, we achieve substantial payload reduction with minimal impact on model fidelity.

3. **Tier 3: Workload Offloading (Retreat Split).** If the constraints (latency $\mathcal{D}$ or memory $\mathcal{M}$) are still not met due to the limited capacity of the front-end device, the algorithm triggers a retreat by selecting a shallower split point (decreasing $\ell$ or $w$). Unlike the previous tiers which focus on compression, this step fundamentally reduces the computational and memory burden on the front-end device by offloading more layers to the back-end server. This serves as a definitive fallback to ensure system feasibility when the edge device cannot handle the processing load.

## 4 EXPERIMENTS

### 4.1 EXPERIMENTAL SETUP

We evaluate our framework on a wide range of datasets, tasks, and backbones. The vision task includes MobileNet-V2 Sandler et al. (2018), ResNet He et al. (2016), CaiT Touvron et al. (2021), LSTR Xu et al. (2021), and YOLOv6s Li et al. (2022) trained on CIFAR-100 Krizhevsky (2009), ImageNet-2012 Deng et al. (2009), and COCO Lin et al. (2014). For NLP, we use Llama 2 Touvron et al. (2023), RoBERTa Liu et al. (2019), QWen2.5 Yang et al. (2024) and Deepseek Shao et al. (2024) evaluated on GLUE Wang et al. (2018), HellaSwag (HS) Zellers et al. (2019), PIQA Bisk et al. (2020), ARC-E/C Clark et al. (2018), GSM8K Cobbe et al. (2021) and HumanEvalChen et al. (2021). Action recognition is evaluated on LSTR on THUMOS'14 Jiang et al. (2014). Unless otherwise specified, we set $\varepsilon = 0.001$, $W = 10\,\mathrm{MHz}$, $\sigma_h^2 = 1$, $\gamma = 10$, $\delta = 0.01$, and $\lambda = 0$. All reported latencies are dataset-averaged values in milliseconds. We employ multi-level quantization: $Q = [q_1, \ldots, q_M]$ splits each IF into $M$ sub-tensors and quantizes them with $q_1 > \ldots > q_M$ bits assigned in descending order of activation magnitude.

### 4.2 RESULTS AND DISCUSSION

Fig. 2 compares our framework against three competing model partitioning (MP) schemes. For vision inference (left), existing methods like Yun et al. Yun et al. (2022) and Hossain et al. Hossain et al. (2023) are limited by bulky intermediate features or restricted split points. This creates a server bottleneck, causing throughput to degrade as more devices are added. In contrast, our framework's fully flexible split point selection enables superior offloading to the front-end, sustaining the highest back-end throughput. This scalability is mirrored in autoregressive inference (right), where our approach reduces the total server (dev$_0$) running time by up to $4.4\times$ compared to the cloud-only EC baseline.

Table 1 (left) shows that for vision tasks, our method achieves a state-of-the-art trade-off. It compresses ResNet-50 features to just 0.08 BPP—an order of magnitude smaller than most prior work—while losing a negligible 0.07 percentage points of top-1 accuracy, proving its efficiency even in fixed bottleneck architectures.

Table 1 (right) compares our framework against lightweight LLM techniques. On Llama-2-7B, our method cuts latency from 7.35s to 1.03s while staying within 2% of baseline accuracy. In contrast, OmniQuant requires 2.5x more bandwidth for similar accuracy, while Atom's accuracy drops by

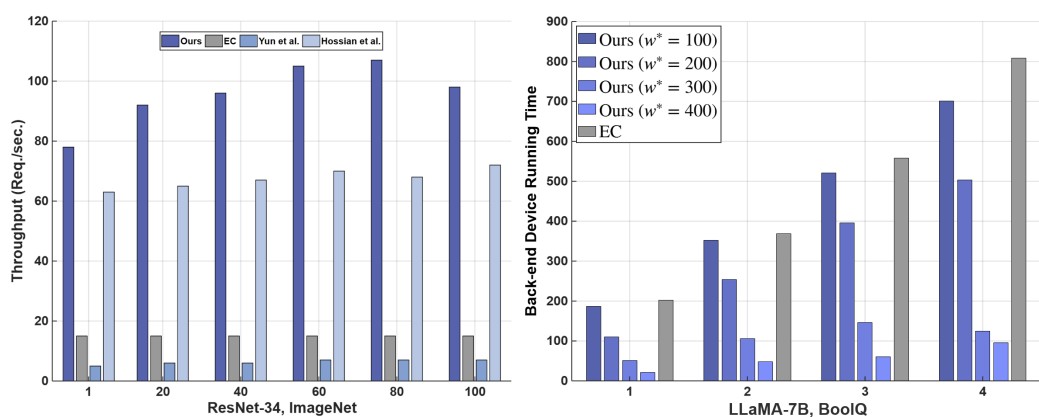

Figure 2: Scalability of our framework in multi-device scenario. Left : Single-shot inference showing the back-end throughput for ResNet-34 on ImageNet as the number of front-end devices increases. Right : Cumulative $dev_0$ running time required to complete the same BoolQ workload with Llama2-7B, compared against the number of front-end devices.

Table 1: (left) Comparison of IF compression methods on ResNet50 (ImageNet). (right) Comparison on LLM lightweight methods on Llama 2-7B. BPP = bits-per-pixel (lower is better); $\Delta$Acc = accuracy change versus the uncompressed baseline.

| Method | $\Delta$Acc | BPP $\downarrow$ |
|---|---|---|
| Matsubara *et al.* | 0.00 | 1.28 |
| Duan *et al.* (n4) | $-0.30$ | 0.70 |
| Duan *et al.* (n8) | $-0.63$ | 0.69 |
| Hossain *et al.* (Cfg-1) | 0.00 | 0.75 |
| Hossain *et al.* (Cfg-2) | 0.46 | 0.69 |
| Ahuja *et al.* | 0.00 | 0.09 |
| **Ours** | $-0.07$ | **0.08** |

| Method | Q | PIQA | ARC-e | ARC-c | HS | $L_c$(ms) |
|---|---|---|---|---|---|---|
| Baseline | 16 | 77.09 | 53.07 | 41.04 | 72.16 | 7352 |
| OmniQuant | 6 | 74.05 | 51.70 | 38.77 | 68.59 | 2504 |
| Atom | 8 | 76.06 | 55.13 | 40.27 | 71.30 | 3576 |
| Atom | 3 | 69.86 | 46.76 | 34.73 | 59.30 | 968 |
| **Ours** | C1 | 75.35 | 54.76 | 39.76 | 69.95 | **1824** |
| **Ours** | C2 | 71.71 | 52.19 | 39.16 | 67.45 | **1032** |

5-10% to match our bandwidth. Our approach is thus unique in its ability to drastically reduce communication without sacrificing task performance.

To strictly validate the server-light design philosophy of SLICER, we conducted a comparative evaluation against two representative split-computing frameworks: BottleFit Matsubara et al. (2022a) and FrankenSplit Furtuanpey et al. (2024) using the ResNet-152 backbone. As summarized in Table 2, existing methods exhibit a severe server-heavy imbalance, offloading only negligible computation (0.97%–3.43% of FLOPs) to the edge. Consequently, their server-side memory footprint surges under concurrency (e.g., BottleFit consumes >600 MB at 50 clients). In contrast, SLICER ($\ell^*$=4) effectively shifts **99.97%** of the computational burden to the edge. This aggressive offloading drastically reduces the peak server memory usage to **55.10 MB**, representing a $10\times$ **reduction** compared to BottleFit. Furthermore, unlike baselines that require extensive retraining yet suffer from accuracy degradation (e.g., -6.58 pp for BottleFit), SLICER maintains near-lossless accuracy (**-0.002 pp**) in a completely training-free manner.

Fig. 3 (left) shows that the overhead from MP and IF compression remains minimal compared to other delay components, and communication overhead is substantially reduced. Moreover, as shown in Fig. 3 (right), our compression method maintains a nearly constant overhead independent of the actual IF size, validating the efficiency of iterative quantization used in *ABQ*.

Table 3 shows that our method delivers strong zero-shot accuracy across every workload, regardless of task type or model family. It remains robust under single-shot inference and—remarkably—handles AR inference just as well, even on demanding code-reasoning and math benchmarks. It even improves

Table 2: Comparison with communication-aware inference frameworks on ResNet-152 (ImageNet).

| Method | Retraining Required? | Top-1 Acc. (%) | $\Delta$ Acc. (pp) ↑ | Edge FLOPs Ratio (%) ↑ | Server Memory (@ 50 Clients) ↓ |
|---|---|---|---|---|---|
| **Baseline (ResNet-152)** | N/A | 78.33 | 0.00 | 100.00 | N/A |
| **BottleFit** Matsubara et al. (2022a) | Yes | 71.73 | -6.58 | 0.97 | 600.95 MB |
| **FrankenSplit** Furtuanpey et al. (2024) | Yes | 77.91 | -0.40 | 3.43 | > 600 MB[†] |
| **SLICER** ($\ell^*$=2) | No | 77.39 | -0.92 | 23.54 | 519.24 MB |
| **SLICER** ($\ell^*$=3) | No | 78.20 | -0.11 | 92.97 | 218.29 MB |
| **SLICER** ($\ell^*$=4) | No | **78.31** | **-0.002** | **99.97** | **55.10 MB** |

[†] Estimated based on trend at 40 clients (572.61 MB).

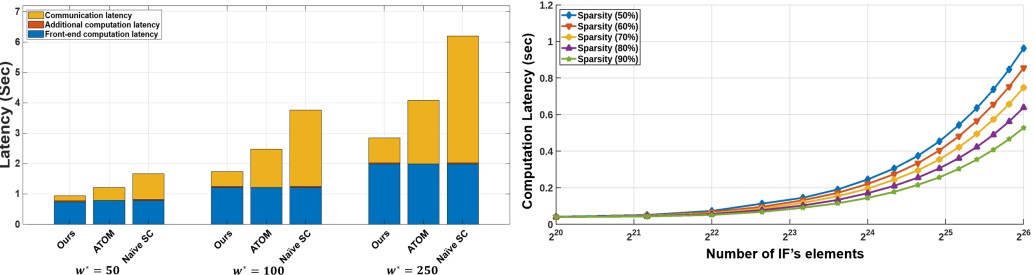

Figure 3: (left) Execution time on the front-end device, IF transmission time, and overhead for the Llama2-7B (BoolQ), comparing different methods across varying device-side computation levels. (right) Computation latency of our framework according to the size of IF.

Table 3: (left) Single-shot inference. (right) AR inference on diverse models and datasets.

| Data | Model | $\ell^*$ | Sparsity | Acc./mAP |
|---|---|---|---|---|
| CIFAR-100 | ResNet34 | 2 | 75% | 72.98(-1.46) |
| ImageNet | ResNet50 | 3 | 75% | 73.01(-1.54) |
| | ResNet50 | 4 | 90% | 74.04(-0.41) |
| | CaiT-XXS24 | 30 | 90% | 81.02(-0.94) |
| GLUE | RoBERTa | 2 | 65% | 90.63(+0.03) |
| | RoBERTa | 18 | 95% | 88.56(-2.04) |
| HellaSwag | Llama2-7B | 3 | 80% | 72.60(-2.20) |
| | Llama2-7B | 20 | 55% | 73.19(-0.71) |
| THUMOS | LSTR | 50 | 55% | 69.60(+0.31) |
| COCO | YOLOv6s | – | 95% | 0.431(-0.019) |

| Data | Model | $w^*$ | Sparsity | Acc. |
|---|---|---|---|---|
| GSM8K (Math) | DeepSeek (Math-7B) | 100 | 70% | 81.05(-0.75) |
| | | 100 | 90% | 81.20(-0.60) |
| GSM8K (Math) | DeepSeek (Math-7B) | 200 | 90% | 81.65(-0.15) |
| | | 200 | 95% | 80.44(-1.36) |
| GSM8K (Math) | Llama3-8B (Instruct) | 100 | 70% | 64.14(-0.61) |
| | | 200 | 90% | 63.76(-0.99) |
| HumanEval (Coding) | Qwen2.5 (Coder-7B) | 100 | 70% | 82.32(+0.61) |
| | | 200 | 70% | 82.93(+1.22) |
| HumanEval (Coding) | Llama3-8B (Instruct) | 100 | 70% | 54.88(+1.83) |
| | | 200 | 90% | 51.83(-1.22) |

accuracy for some tasks, which is a common phenomenon across most tasks and is due to the noise removal effect in IF.

Table 4: **Multi-modal (text-to-image) generation.**: *FLUX.1-dev* (with Realism LoRA) and *AIDC-AI/Ovis-U1-3B*. **Baseline** offloads the entire generation to the back-end with no partitioning/compression. **Ours** applies our method at diffusion timestep $t$ with sparsity $s \in \{0.6, 0.8\}$. Quantization bit-width $Q = [8, 8, 8]$; ABQ threshold $\delta = 0.2$.

| **FLUX.1-dev + RealismLoRA** | | | | |
|---|---|---|---|---|
| Sparsity | Timestep $t$ | PSNR ↑ | SSIM ↑ | CLIP-IQA ↑ |
| EC (Baseline) | – | 16.61 ± 4.20 | 0.76 ± 0.12 | 0.76 ± 0.17 |
| 0.6 | 50 | 16.91 ± 4.19 | 0.76 ± 0.12 | 0.77 ± 0.15 |
| 0.8 | 50 | 16.62 ± 4.24 | 0.76 ± 0.12 | 0.76 ± 0.15 |
| 0.6 | 20 | 16.81 ± 4.05 | 0.76 ± 0.12 | 0.76 ± 0.17 |
| 0.8 | 20 | 16.93 ± 4.16 | 0.76 ± 0.12 | 0.77 ± 0.15 |

| **AIDC-AI/Ovis-U1-3B** | | | | |
|---|---|---|---|---|
| Sparsity | Timestep $t$ | PSNR ↑ | SSIM ↑ | CLIP-IQA ↑ |
| EC (Baseline) | – | 33.73 ± 4.72 | 0.98 ± 0.01 | 0.92 ± 0.03 |
| 0.6 | 50 | 39.28 ± 4.10 | 0.99 ± 0.01 | 0.92 ± 0.03 |
| 0.8 | 50 | 36.19 ± 3.29 | 0.98 ± 0.01 | 0.92 ± 0.03 |
| 0.6 | 20 | 34.55 ± 5.12 | 0.98 ± 0.01 | 0.92 ± 0.03 |
| 0.8 | 20 | 28.43 ± 5.77 | 0.96 ± 0.02 | 0.92 ± 0.02 |

We evaluated text-to-image diffusion models under aggressive sparsification and quantization. As summarized in Table 4, PSNR/SSIM remain stable on FLUX.1-dev (with Realism LoRA), and CLIP-IQA stays essentially unchanged on AIDC-AI/Ovis-U1-3B, indicating that the semantic alignment

between prompt and image is preserved even at high sparsity. These results validate that our framework applies to complex, iterative, cross-modal pipelines without degrading essential semantic coherence. See Appendix F for output samples.

Table 5: (left) Ablation study for each module. (mid) Preventing quantization distortion and increasing sparsity effects according to MS. (right) Average bit savings by ABQ.

**ATKF −**

| MS | ABQ | Acc. | $L_c$ (ms) |
|----|-----|------|-----------|
| − | − | 74.53 | 9592 |
| + | − | 74.54 | 7928 |
| − | + | 74.54 | 8720 |
| + | + | 74.58 | 7152 |

**ATKF +**

| MS | ABQ | Acc. | $L_c$ (ms) |
|----|-----|------|-----------|
| − | − | 73.02 | 4736 |
| + | − | 73.03 | 3640 |
| − | + | 73.01 | 4304 |
| + | + | 73.01 | **3312** |

**MS**

| $M$ | $Q$ | Acc. | $L_c$ (ms) |
|-----|-----|------|-----------|
| 1 | [8] | 73.02 | 3696 |
| 1 | [8,8] | 73.04 | 4024 |
| 2 | [8,2] | 73.05 | 3232 |
| 3 | [4,4,4] | 72.98 | 3328 |
| 3 | [4,1,1] | 72.98 | 2840 |
| 3 | [1,4,4] | 65.08 | 3176 |
| 3 | [1,4,1] | 65.08 | 2928 |
| 3 | [1,1,4] | 68.10 | 2928 |
| 4 | [8,4,2,1] | 73.01 | 3080 |
| 4 | [1,2,4,8] | 68.08 | 3184 |

**Sparsity = 65%**

| $\delta$ | Acc. (%) | $Avg.Q$ |
|----------|----------|---------|
| 0.05 | 73.20 | [4.0, 7.1, 7.2] |
| 0.10 | 73.21 | [4.0, 7.1, 6.8] |
| 0.15 | 73.17 | [4.0, 6.2, 6.3] |
| 0.20 | 73.19 | [4.0, 5.7, 5.3] |

**Sparsity = 85%**

| $\delta$ | Acc. (%) | $Avg.Q$ |
|----------|----------|---------|
| 0.05 | 69.80 | [4.0, 7.0, 7.0] |
| 0.10 | 69.80 | [4.0, 7.0, 7.0] |
| 0.15 | 69.80 | [4.0, 6.9, 6.9] |
| 0.20 | 69.82 | [4.0, 5.9, 5.8] |

Table 5 (left) presents an ablation study under fixed settings ($\ell = 5$, $s = 0.75$). Activating our proposed modules in sequence (ATKF → MS + ABQ) achieves the best accuracy-latency trade-off, reducing latency by nearly 65% to 3312 ms with an accuracy drop of less than 1.5 percentage points. The middle table validates our magnitude-aware bit allocation (MS). Assigning more bits to the most significant block ($[8, 2]$) maintains an accuracy of 73.05%, whereas reversing the allocation ($[1, 4, 4]$) causes a sharp drop to 65.08% for a similar latency. Finally, the right table shows that as $\delta$ increases, our adaptive bit quantization (ABQ) effectively reallocates an initial $[8, 8, 8]$ budget. This is achieved with a negligible accuracy decrease of less than 0.05 pp at both 65% and 85% sparsity levels.

## 5 CONCLUSIONS

We introduced a novel model partitioning framework to mitigate communication and server-side bottlenecks in deploying large-scale DNNs on multi device setup. By selectively discarding low-magnitude activations and assigning appropriate bitwidths to larger ones, we substantially reduce bandwidth usage with minimal impact on model accuracy. Experimental results on various benchmarks confirm that our framework consistently lowers E2E latency and scales well under multi-device concurrency. Interestingly, its lightweight design does not necessitate retraining or structural modifications, enabling seamless integration with a wide spectrum of DNN architectures, including AR inference in LLMs. Our framework provides a versatile platform for distributed inference, enabling scalable, adaptive, and efficient AI deployments in real-world scenarios.

## REPRODUCIBILITY STATEMENT

We have made every effort to ensure our work is reproducible. The complete implementation of the **SLICER** framework, including source code, configuration files, and reproducibility scripts, is provided in the supplementary material. Details of the experimental setup, including model architectures, benchmark datasets, and hyper-parameters, are described in Section 4.1. The software/hardware environment and the rationale for each setting are further detailed in Appendix G.1. The core components of our compression pipeline—ATKF, MS, and ABQ—are explained in Section 3. For clarity, pseudo-code for the ENCODE, DECODE, and SLICER-SEARCH algorithms is provided in Appendices E-F (Algorithms 1-3), and a step-by-step numerical example of the compression process is included in Appendix A. Comprehensive ablation studies and additional qualitative/quantitative results are available in Section 4.2 and Appendix F. We are confident that these materials ensure the transparency of our findings and enable the independent verification and faithful reproduction of our results.

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
