# OpenReview forum: "Why Should the Server Do It All?: A Scalable, Versatile, and Model-Agnostic Framework for Server-Light DNN Inference over Massively Distributed Clients via Training-Free Intermediate Feature Compression"
_ICLR.cc/2026/Conference — Submitted to ICLR 2026_

### Official Review · Reviewer_Q13w · 2025-10-29

**Soundness:** 3
**Presentation:** 3
**Contribution:** 2
**Rating:** 4
**Confidence:** 3

**Summary:**

This paper proposes SLICER, a training-free, model-agnostic IF codec that attaches to modern DNN
models and reduces uplink volume and server load. The contributions are:
(i) Asymmetric Top-K Filtering (ATKF) to enforce exact sparsity,
(ii) Magnitude-Splitting (MS) into equal-cardinality blocks, and
(iii) Adaptive Bit Quantization (ABQ) with an integer-only distortion guard.
Reported gains include up to 10x uplink reduction and 4.4x server time speedup with little accuracy drop across vision and LLM tasks.

**Strengths:**

1. It identifies a key problem in edge-cloud DNN inference and proposes a novel, training-free IF compression framework.

2. The proposed techniques (ATKF, MS, ABQ) are well-motivated and effectively reduce uplink volume and server load.

3. Scales in multi-device settings and is model-agnostic, making it versatile for various applications.

**Weaknesses:**

1. There is no analysis on the long token length setting for LLMs with SLICER, which maybe impact by the proposed ATKF method.

2. There is no device-side profiling for computation and memory overhead.

3. Some varibales seems arbitrary, such as the choice of equal block size in MS.

**Questions:**

Overall, I think this paper address an important problem in the realm of edge-cloud computing, and the idea is interesting. I have a few comments and questions as follow.
Citation Error: There should be a citation for Llama 2 in the section 4.1.

1. How does quantization/sparsity noise in IFs propagate over long token generations in LLMs?

2. What are measured codec compute/energy costs on representative edge SoCs, and how do they scale with IF size/depth?

3. Does ATKF random tie-breaking induce nondeterminism affecting reproducibility or accuracy?

4.  The hierarchical search adjusts sparsity/split under budgets, but stability behavior under fluctuating networks isn’t deeply analyzed.

5. Are features compressed per layer, per token, or globally across the model? How does the granularity influence both latency and reconstruction fidelity?

---

> ### Author Response · Authors · 2025-11-25
>
> # [Response to Q13w. Weakness 1 / Question 1]
>
>
> Thank you for very much for the comment. In the revised version, we have included comprehensive analysis for the long token length. To this end and to investigate whether ATKF-induced quantization noise accumulates over long token generations, we have conducted additional experiments on HumanEval and GSM8K with prompts extended up to 4096 tokens. The results have been reported in Table 14 of Appendix F in the revised manuscript and demonstrate that:
>
> - Our framework (SLICER) still maintains strong robustness without catastrophic degradation, even in long-context scenarios.
>
> - For Llama3-8B, it consistently achieves higher accuracy than the baseline, while for Qwen2.5, it remains stable within a negligible margin.
>
> - Although it shows minor sensitivity for DeepSeek-Math at the extreme 4096-token length, the overall trend confirms that our compression approach effectively preserves reasoning capabilities across extended sequences.
>
> | Model / Benchmark | 300 / 512 | 400 / 1024 | 500 / 2048 | 600 / 4096 |
> | :--- | :---: | :---: | :---: | :---: |
> | **Llama3-8B** / HumanEval | 58.54% $\color{red}{(+5.49)}$ | 59.76% $\color{red}{(+6.71)}$ | 57.93% $\color{red}{(+4.88)}$ | 56.71% $\color{red}{(+3.66)}$ |
> | **Qwen2.5** / HumanEval | 79.88% $\color{blue}{(-1.83)}$ | 79.27% $\color{blue}{(-2.44)}$ | 79.88% $\color{blue}{(-1.83)}$ | 81.71% (0.00) |
> | **DeepSeek-Math** / GSM8K | 82.41% $\color{red}{(+0.61)}$ | 82.03% $\color{red}{(+0.23)}$ | 82.18% $\color{red}{(+0.38)}$ | 71.11% $\color{blue}{(-10.69)}$ |
> | **Llama3-8B** / GSM8K | 66.11% $\color{red}{(+1.36)}$ | 65.81% $\color{red}{(+1.06)}$ | 66.72% $\color{red}{(+1.97)}$ | 68.23% $\color{red}{(+3.48)}$ |
>
> ---
>
> # [Response to Q13w. Weakness 2 / Question 2]
>
> Thank you very much for the comment. In the revised version, we have included analysis on device-side profiling for computation and memory overhead and on scaling with IF size/depth.
> To this end, we have expanded the experimental evaluation by including comprehensive measurements on a diverse array of representative edge SoCs: Jetson TX1, Xavier NX, Orin NX, AGX Orin, and reference desktop GPUs (RTX 2060, 3060). The results have been provided in Figure 7 of Appendix F and confirm that the overhead is minimal and scales efficiently, as detailed below.
>
> - **Computation Overhead and Scaling:** The codec overhead scales linearly with IF size. Even on the most resource-constrained device (Jetson TX1), the encoding latency is approx. 49 ms for a standard ResNet workload (1.5 MB tensor), which is negligible compared to the transmission latency savings (>2,000 ms gain on 2MHz bandwidth). As the split point deepens, the spatial resolution decreases, further reducing the encoding burden in single-shot models.
>
> - **Energy Efficiency:** We have measured the energy cost per inference. The additional energy consumed by the SLICER codec (approx. 267 mJ on Jetson TX1) is significantly outweighed by the reduction in wireless transmission energy. Considering the 82\% data reduction, the system saves over 2,000 mJ in transmission energy (assuming standard WiFi Tx power), confirming that SLICER extends battery life on edge devices rather than draining it.
>
> - **Scaling on split depth:** We distinct trends across modalities: in single-shot inference, the IF size generally decreases as the split point deepens due to spatial downsampling, whereas in Autoregressive (AR) inference, the effective data volume typically increases or accumulates as the sequence length grows.

---

> ### Author Response · Authors · 2025-11-25
>
> # [Response to Q13w. Weakness 3]
>
> Thank you very much for the important comment. In the revised version, we have improved explanations on the hyperparameter (or variable) selection and clarified how key hyperparameters were selected.
> In our framework, the hyperparameters are categorized into two distinct groups: (1)  **Empirically Tuned Hyperparameters** ($\delta, \lambda$), which are calibrated based on task characteristics, and (2)  **Adaptively Determined Hyperparameters** ($s, M$), which are automatically optimized by our mechanism.
>
>  **1. Empirically Tuned Hyperparameters ($\delta, \lambda$)**
> These hyperparameters determine the rate-distortion behavior and are selected to match the specific sensitivity of the target task.
>
> -  **Distortion Threshold $\delta$:** This hyperparameter controls the aggressiveness of the Adaptive Bit Quantization (ABQ). As analyzed in Table 10 (Appendix F), $\delta$ allows users to trade off between bit-width reduction and accuracy maintenance. In the experiments, although we select $\delta=0.01$ as (robust) default via empirical validation, its value could be tuned in practice to maximize the compression for tasks with higher error tolerance.
>
> -  **Asymmetry Factor $\lambda$:** This parameter controls the cut-off thresholds for positive and negative activations in ATKF. $\lambda$ is deemed as a task-specific hyperparameter to be tuned for optimal model fidelity.
> As shown in Table 11 (Appendix F), the optimal $\lambda$ is highly task-dependent. In our framework, we select negative values of $\lambda$ (e.g., $\lambda = -0.5$) to significantly improve performance on reasoning tasks like PIQA and Wino, whereas we select $\lambda=0$ for other tasks.
>
>  **2. Adaptively Determined Hyperparameters ($s, M$)**
> Unlike the empirically tuned hyperparameters, the sparsity level ($s$) and block count ($M$) are not manually selected. These hyperparameters are adaptively determined by the proposed SLICER-SEARCH method (Algorithm~3) to fulfill the constraints.
>
> -  **Automated Mechanism:** Under the system constraints (latency budget $\mathcal{D}$ and memory budget $\mathcal{M}$) and given the set of empirically determined hyperparameters $\theta=\{\delta, \lambda\}$, SLICER-SEARCH automatically finds the optimal configuration of $s$ and $M$.
>
> -  **Search Policy:** The mechanism of SLICER-SEARCH is based on a hierarchical search policy to minimize accuracy loss. It first adjusts the block granularity ($M$) to optimize the header overhead, and then, increases sparsity ($s$) or retreats the split point only if necessary to meet the hard constraints. This ensures reproducibility and eliminates the need for manual intervention during inference.
>
> ---
>
> # [Response to Q13w. Question 3]
>
>
> To answer the question and to rigorously quantify the relevant effect, in the revised version, we have conducted an additional empirical analysis specifically targeting the nondeterminism mechanism. We have performed 30 repeated experiments for each model, keeping all hyperparameters and settings identical, except for the random seed governing the ATKF tie-breaking process.
>
> - The table below presents the mean and standard deviation of the HumanEval accuracy across these 30 runs for both Llama-3 and Qwen2.5, under various sparsity ($s$) levels:
>
> | Model | $s=0.2$ | $s=0.5$ | $s=0.7$ | $s=0.9$ |
> | :--- | :---: | :---: | :---: | :---: |
> | **Llama-3**|$54.70 \pm 1.99$|$53.36 \pm 1.08$|$53.17 \pm 1.21$|$52.87\pm0.73$|
> | **Qwen2.5**|$81.65 \pm 0.93$|$82.08 \pm 1.68$|$83.11 \pm 1.73$|$82.75 \pm1.47$|
>
> - As the results demonstrate, the standard deviation is consistently low across all models and sparsity settings. This low variance indicates that while the random tie-breaking introduces a minor stochastic element, its impact on the final model accuracy is negligible. Therefore, it does not significantly affect the reproducibility or stability of our framework.

---

> ### Author Response · Authors · 2025-11-25
>
> # [Response to Q13w. Question 4]
>
> Thank you very much for the comment. To address the concern, in the revised version, we have included in-depth analysis on the stability behavior under fluctuating networks.
> To this end, we have conducted additional experiments by varying channel bandwidths (10, 5, 2 MHz) under fixed split conditions. The results have been reported in Table 15 of Appendix F.
>
> - Our hierarchical search still demonstrates the robust adaptation through SLICER-SEARCH, a mechanism that optimizes the configuration of MS, sparsity, and the split layer to align with strict latency and memory budgets.
>
> - Particularly, by adaptively tuning these parameters (MS, sparsity, and the split layer) according to the network fluctuations and system constraints, the performance of SLICER is still maintained within tolerable or acceptable ranges, albeit at the potential cost of slight degradation. It also maintains the system feasibility and high accuracy even under severe bandwidth constraints, validating the stability and robustness of our approach.
>
> $$
> \\begin{array}{c|ccc|c}
> \\hline
> \\textbf{Sparsity} & \\textbf{10 MHz} & \\textbf{5 MHz} & \\textbf{2 MHz} & \\textbf{Acc.} \\\\
> & \\text{(ms)} & \\text{(ms)} & \\text{(ms)} & \\\\
> \\hline
> 0\\% & 724 & 1098 & 2055 & 74\\% \\\\
> 5\\% & 689 & 1045 & 1956 & 74\\% \\\\
> 15\\% & 621 & 942 & 1764 & 74\\% \\\\
> 25\\% & 554 & 840 & 1572 & 74\\% \\\\
> 35\\% & 484 & 734 & 1375 & 74\\% \\\\
> 45\\% & 415 & 630 & 1180 & 74\\% \\\\
> 55\\% & 346 & 525 & 983 & 73\\% \\\\
> 65\\% & 277 & 420 & 785 & 72\\% \\\\
> 75\\% & 206 & 312 & 584 & 71\\% \\\\
> \\hline
> \\end{array}
> $$
>
> ---
>
> # [Response to Q13w. Question 5]
>
>
> Thank you very much for the comment. In the revised version, we have clarified that in our framework, the feature compression is performed at a particular layer, as detailed below.
>
> - Our framework (SLICER) operates on a Layer-wise Global Granularity, which is further refined into Magnitude-Based Partitions.
>
> - Specifically, in our framework, we flatten the entire intermediate feature tensor and sort non-zero values by magnitude, grouping them into a small number of blocks ($M$, typically $2\sim4$). This strategy enables optimizing the trade-off between fidelity and latency.
>
>
> Also, in the revised version, we have further explored how the granularity influences both latency and reconstruction fidelity, as detailed below.
>
> - **Reconstruction Fidelity and Magnitude Adaptation:** By grouping similar values (e.g., large vs. small), each partition calculates its own min/max range. This prevents outliers in the **large value block** from destroying the precision of small value blocks, significantly reducing quantization error compared to a single layer-wide scale.
>
> - **Full-Range Utilization:** Within each partition, we employ Asymmetric Quantization. Unlike symmetric schemes that often waste half the quantization range (e.g., unused negative bins for positive-only data), our approach maps the partition's exact [min, max] to the full integer range (e.g., $[0, 2^q-1]$), ensuring 100\% bin utilization and higher fidelity.
>
> - **Communication Latency and Minimal Metadata:** Per-token or per-channel granularity requires transmitting scale/zero-point parameters for every unit, causing significant overhead. In contrast, SLICER only requires metadata for the $M$ partitions. This structurally negligible overhead ensures that our high-fidelity compression does not become a communication or computational bottleneck.
>
> - **Computation Latency:** The chosen granularity keeps encoder-side cost low. ATKF, MS, and ABQ are implemented as a single global sort followed by a few linear passes over the tensor, with per-block bit search done in integer arithmetic. As shown in the latency measurements (Table 7 of Appendix F), the encode time remains a small, nearly constant fraction of end-to-end latency even when the IF size doubles, whereas finer per-token or per-channel granularity would require many small kernels and extra metadata.

---

### Official Review · Reviewer_FSvt · 2025-11-01

**Soundness:** 2
**Presentation:** 3
**Contribution:** 3
**Rating:** 4
**Confidence:** 4

**Summary:**

Model partitioning in client–server architectures for DNN inference typically relies on static split points, which can overload the server and underutilize edge devices. This issue is especially critical for large language models (LLMs), where autoregressive generation produces large intermediate features (IFs). The paper introduces SLICER, an architecture-agnostic framework that splits models and compresses IFs to reduce server load and communication costs in split computing. SLICER employs a three-stage pipeline (Asymmetric Top-K Filtering (ATKF), Magnitude Splitting (MS), and Adaptive Bit Quantization (ABQ)) to determine split points that balance latency, memory, and bandwidth constraints. The experiments demonstrate the framework’s effectiveness across different domains.

**Strengths:**

+ The method is validated on multiple domains, showing consistent performance improvements.
+ The authors have released the implementation, promoting transparency and facilitating future research.

**Weaknesses:**

- The paper does not compare SLICER against prior split-computing or communication-aware inference frameworks, such as BottleFit [1] and Frankensplit [2]. Including these baselines would better suit SLICER’s contributions.
- The experiments use only a single client device (an NVIDIA Jetson AGX Orin) in the client–server pipeline. This setup is too limited to substantiate claims about scalability and generalization to multiple clients.
- The hyperparameter selection process (e.g., sparsity level s, distortion threshold δ, block count M, asymmetry factor λ) is not well explained, limiting reproducibility and interpretability.
- The discussion focuses solely on single-point splitting. It would be beneficial to explore the implications and challenges of multi-point partitioning.
- Although multi-client scenarios are briefly mentioned, the paper does not discuss whether synchronization mechanisms are required when multiple front-end devices are involved.

Reference:

[1] Matsubara, Yoshitomo, Davide Callegaro, Sameer Singh, Marco Levorato, and Francesco Restuccia. "Bottlefit: Learning compressed representations in deep neural networks for effective and efficient split computing." In 2022 IEEE 23rd International Symposium on a World of Wireless, Mobile and Multimedia Networks (WoWMoM), pp. 337-346. IEEE, 2022.

[2] Furutanpey, Alireza, Philipp Raith, and Schahram Dustdar. "Frankensplit: Efficient neural feature compression with shallow variational bottleneck injection for mobile edge computing." IEEE Transactions on Mobile Computing 23, no. 12 (2024): 10770-10786.

**Questions:**

1. Since SLICER is a partitioning framework, would it be feasible to include direct comparisons with existing split-computing frameworks such as BottleFit or Frankensplit?
2. The experimental setup involves only one client device. Could the authors expand the evaluation to include multiple or heterogeneous edge devices to assess scalability better?
3. How were key hyperparameters (e.g., s, δ, M, λ) selected? Were they tuned empirically, or is there a mechanism to adaptively determine them in practice?
4. The paper discusses partitioning at a single split point. Is it possible to extend SLICER to support multiple split points within a DNN? If so, how might this impact performance and communication costs?
5. The “Hierarchical Search Policy” (lines 324–327) is mentioned briefly. Could the authors provide a more detailed explanation of how this policy operates and influences split-point determination?

---

> ### Author Response · Authors · 2025-11-25
>
> # [Response to FSvt. Weakness 1 / Question 1]
>
> We sincerely thank the reviewer for the valuable suggestion.
>
> - In the revised version, we have conducted additional experiments and comparative study with prior frameworks, BottleFit [1] and FrankenSplit [2], to further validate the efficacy and supremacy of our framework against [1] and [2]. The experimental results have been reported in Table 2 of the revised manuscript. Here, the Edge FLOPs Ratio (a performance metric) denotes the proportion of the total model FLOPs that are executed on the edge device, which measures how actively the edge resources are utilized.
>
> - The results clearly demonstrates that SLICER achieves a superior trade-off compared to the prior methods [1] and [2]. It is because, unlike [1] and [2] that underutilized the edge devices' resources and offloaded the majority of workloads to the server, SLICER actively utilizes edge devices' resources. This in turn enables significantly reducing the server operating expenses while maintaining the near-lossless accuracy, thereby validating the effectiveness of our approach as a scalable, training-free solution.
>
> | Method | Retrain? | Top-1 Acc.(%) | $\Delta$ Acc.(pp) $\uparrow$ | Edge FLOPs Ratio (%) $\uparrow$ | Server Mem.(@ 50 Clients) $\downarrow$ |
> | :--- | :---: | :---: | :---: | :---: | :---: |
> | **Baseline** | N/A | 78.33 | 0.00 | 100.00 | N/A |
> | BottleFit | Yes | 71.73 | -6.58 | 0.97 | 600.95 MB |
> | FrankenSplit | Yes | 77.91 | -0.40 | 3.43 | $> 600$ MB$^\dagger$ |
> | **SLICER ($\ell^{\ast}$=2)** | **No** | 77.39 | -0.92 | 23.54 | 519.24 MB |
> | **SLICER ($\ell^{\ast}$=3)** | **No** | 78.20 | -0.11 | 92.97 | 218.29 MB |
> | **SLICER ($\ell^{\ast}$=4)** | **No** | **78.31** | **-0.002** | **99.97** | **55.10 MB** |
>
> ---
>
> # [Response to FSvt. Weakness 2 / Question 2]
>
>
> We appreciate the Reviewer's comment on the experimental setup.
>
> - To address the concern about scalability and generalization, in the revised version, we have expanded our evaluation and experiment to a realistic heterogeneous multi-client environment. Specifically, we have constructed an additional testbed consisting of four distinct edge devices with diverse computational capabilities and memory constraints: RTX 3060 (6GB, INT4), RTX 2060 Super (8GB, INT8), RTX 3080 (12GB, INT8), and Jetson AGX Orin (32GB, FP16), all connected to a single NVIDIA A6000 server. We have utilized the CNN/DailyMail (Version 3.0.0) dataset for the evaluation.
> The experimental results along with detailed and thorough discussions have been provided in Table 13 of Appendix F in the revised manuscript.
>
> - The experimental results demonstrate that our framework still excels in adapting the split point according to multiple devices' specific (and different) hardware constraints. Particularly, as more powerful edge devices are involved, our framework effectively reduces the server's overhead by offloading a larger portion of computation to the client side.
> SLICER achieves 1.60$\times$ speedup and 43.9\% reduction in the server workload compared to the baselines, thereby confirming load balancing and robust performance of our approach across heterogeneous client setups.
>
> | Scenario | Latency (s) $\downarrow$ | Throughput $\uparrow$ | Server Tokens $\downarrow$ | Reduction (%) $\uparrow$ | Speedup $\uparrow$ |
> | :--- | :---: | :---: | :---: | :---: | :---: |
> | Baseline | 364.21 | 39.5 | 1,076,161 | - | 1.00$\times$ |
> | Scenario 1 | 320.41 | 44.9 | 928,291 | 13.7% | 1.14$\times$ |
> | Scenario 2 | 227.58 | 63.3 | 604,173 | 43.9% | 1.60$\times$ |
> | **Scenario 3** | **97.08** | **148.3** | **172,400** | **84.0%** | **3.75$\times$** |
>
> *Scenario Configs: **S1** (4$\times$3060), **S2** (1$\times$3060, 1$\times$2060S, 1$\times$3080, 1$\times$Orin), **S3** (2$\times$3080, 2$\times$Orin)*
>
> ---

---

> ### Author Response · Authors · 2025-11-25
>
> # [Response to FSvt. Weakness 3 / Question 3]
>
>
> Thank you very much for the important comment. In the revised version, we have improved explanations on the hyperparameter selection and clarified how key hyperparameters were selected.
> In our framework, the hyperparameters are categorized into two distinct groups: (1) **Empirically Tuned Hyperparameters** ($\delta, \lambda$), which are calibrated based on task characteristics, and (2) **Adaptively Determined Hyperparameters** ($s, M$), which are automatically optimized by our mechanism.
>
> **1. Empirically Tuned Hyperparameters ($\delta, \lambda$)**
> These hyperparameters determine the rate-distortion behavior and are selected to match the specific sensitivity of the target task.
>
> - **Distortion Threshold $\delta$:** This hyperparameter controls the aggressiveness of the Adaptive Bit Quantization (ABQ). As analyzed in Table 10 (Appendix F), $\delta$ allows users to trade off between bit-width reduction and accuracy maintenance. In the experiments, although we select $\delta=0.01$ as (robust) default via empirical validation, its value could be tuned in practice to maximize the compression for tasks with higher error tolerance.
>
> - **Asymmetry Factor $\lambda$:** This parameter controls the cut-off thresholds for positive and negative activations in ATKF. $\lambda$ is deemed as a task-specific hyperparameter to be tuned for optimal model fidelity.
> As shown in Table 11 (Appendix F), the optimal $\lambda$ is highly task-dependent. In our framework, we select negative values of $\lambda$ (e.g., $\lambda = -0.5$) to significantly improve performance on reasoning tasks like PIQA and Wino, whereas we select $\lambda=0$ for other tasks.
>
> **2. Adaptively Determined Hyperparameters ($s, M$)**
> Unlike the empirically tuned hyperparameters, the sparsity level ($s$) and block count ($M$) are not manually selected. These hyperparameters are adaptively determined by the proposed SLICER-SEARCH method (Algorithm 3) to fulfill the constraints.
>
> - **Automated Mechanism:** Under the system constraints (latency budget $\mathcal{D}$ and memory budget $\mathcal{M}$) and given the set of empirically determined hyperparameters $\theta=\{\delta, \lambda\}$, SLICER-SEARCH automatically finds the optimal configuration of $s$ and $M$.
>
> - **Search Policy:** The mechanism of SLICER-SEARCH is based on a hierarchical search policy to minimize accuracy loss. It first adjusts the block granularity ($M$) to optimize the header overhead, and then, increases sparsity ($s$) or retreats the split point only if necessary to meet the hard constraints. This ensures reproducibility and eliminates the need for manual intervention during inference.
>
> ---
>
> # [Response to FSvt. Weakness 4 / Question 4]
>
>
> We appreciate the important comment and valuable suggestion. In the revised version, we have thoroughly explored the multi-point partitioning in various aspects.
> To this end, we have conducted additional experiments by extending our framework (SLICER) to enable multiple split points (from 1 to 4) and evaluating the performance on the Llama3-8B model using the HumanEval benchmark. The results have been included in Table 16 of Appendix F.
>
> - **Impact on Performance:**
> As observed from the results of Table 16, increasing the number of split points leads to a slight degradation in accuracy. This is primarily due to the accumulation of quantization errors. Specifically, since features are compressed and restored at each split point, the reconstruction noise tends to compound and accumulate as the number of partitions increases. However, it is important to see that this performance drop is marginal. Even with up to 4 split points, the model maintains competitive accuracy, demonstrating the robustness and versatility of SLICER for effectively handling the multi-point splitting with no critical failure.
>
> - **Communication Costs and System Implications:**
> Simply increasing the number of split points does not inherently reduce the total communication cost. In fact, in practical (noisy) wireless environments, multiple transmission events can exacerbate the latency due to accumulated overheads (e.g., connection establishment and retransmissions), even when the payload is significantly compressed by our framework.
>
> - **Challenges:** We found that optimizing a scheduling strategy for multi-point partitioning is an intriguing, yet challenging, avenue for holistic resource utilization across all hops.
> Another challenge is the (complex) trade-off between distributed computation and multi-hop communication overhead. How to overcome these challenges needs to be further explored in the literature.
>
> | Sparsity ($s$) | Split: 1| Split: 2|Split: 3|Split: 4|
> | :--- | :---: | :---: | :---: | :---: |
> | $s=0.1$|53.05|52.44|51.22|50.61|
> | $s=0.3$|51.22|52.44|53.05|50.61|
> | $s=0.5$|54.27|53.05|51.83|51.22|
> | $s=0.7$|52.44|51.83|51.83|50.00|
>
> *Accuracy (%) on HumanEval with varying number of Split Points.*

---

> ### Author Response · Authors · 2025-11-25
>
> # [Response to FSvt. Weakness 5]
>
>
> Thank you very much for the comment. In our framework, no synchronization mechanism is required between front-end devices, which is indeed a significant merit in practice. Specifically, it is due to:
>
> - **Independent \& Stateless Operation:** Each device optimizes compression locally using SLICER-Search and requests asynchronously. The back-end treats incoming IFs as independent jobs (via standard queues or batching) without maintaining shared state or coordination across clients.
>
> - **Decoupled Control Plane:** Also, the updates to the latency target $D_i$ (to reflect server load) occur on a long-term (slow) timescale and do not require lock-step synchronization.
>
> In the revised version, we have explicitly stated in Sec. 2.1 and Sec. 4.2 that front-end devices operate asynchronously and require no device-to-device coordination.
>
> ---
>
> # [Response to FSvt. Question 5]
>
>
> Thank you very much for the comment. In the revised version, we have provided more detailed explanations of how the **Hierarchical Search Policy** operates and its influences on split-point determination.
>
> The **Hierarchical Search Policy** is the core decision logic of SLICER, designed to satisfy latency ($\mathcal{D}$) and memory ($\mathcal{M}$) constraints while minimizing accuracy degradation.
> The policy follows an **Accuracy-First** principle, prioritizing lossless or low-distortion adjustments before resorting to workload shifts. The process operates in three distinct tiers, as described in Algorithm 3 and detailed below.
>
> - **Tier 1: Structural Optimization (Tune $M$).**
> First, the algorithm adjusts the block count ($M$) to optimize the trade-off between quantization granularity and metadata overhead. This step is prioritized as it optimizes the bitstream structure without pruning any activations, ensuring zero impact on model fidelity.
>
> - **Tier 2: Controlled Distortion (Increase $s$).**
> If constraints remain unmet, the policy incrementally increases sparsity ($s$). This introduces controlled distortion by removing low-magnitude values. It is prioritized second because, unlike Tier 1, it trades off a small amount of information for significant compression.
>
> - **Tier 3: Workload Offloading.**
> If the edge device's capacity limits prevent satisfying constraints even with maximum compression, the algorithm selects a shallower split point (decreasing $\ell$ or $w$). Unlike previous tiers, this step reduces the computational and memory burden on the front-end device by offloading more layers to the back-end server. It serves as a definitive fallback to ensure system feasibility.
>
> **Influence on Split-Point Determination:**
> By attempting to compress the IF at the deepest possible layer first (via Tiers 1 \& 2), SLICER sustains server-side offloading longer than baselines. The split point is only retreated (Tier 3) when the edge device cannot handle the processing load, ensuring optimal resource utilization.
>
> We have revised Section 3.3 in the manuscript to explicitly include this detailed breakdown of the prioritization logic.

---

### Official Review · Reviewer_TyTH · 2025-11-01

**Soundness:** 2
**Presentation:** 2
**Contribution:** 3
**Rating:** 4
**Confidence:** 3

**Summary:**

The paper proposes SLICER, a training-free, model-agnostic framework for server-light split computing that compresses intermediate features (IFs) to reduce both uplink traffic and server load. The codec combines Asymmetric Top-K Filtering (ATKF) for exact sparsity control, Magnitude-Splitting (MS) into equal-cardinality blocks, and Adaptive Bit Quantization (ABQ) with a lightweight integer mismatch guard. The method includes a predictive, constraint-aware configuration policy for both single-shot and autoregressive (AR) inference.

**Strengths:**

+The framework seems that it training-free and plug-and-play.
+ The formulation selects (w,ℓ) and compression knobs θ to meet latency/memory caps and stabilize bits-per-token and server time/token—addressing a real bottleneck for multi-client LLM services.
+ Results span CNNs/transformers, LLM AR decoding, and diffusion; multi-device scaling shows server time reductions up to 4.4× and sustained backend throughput as clients grow; vision achieves state-of-the-art BPP for IFs with negligible accuracy loss.

**Weaknesses:**

- The concrete edge device in this paper used is never mentioned in this paper. As such, the latency measurement is unfair since latency depends on the computation ability of the concrete device.

- The evaluation largely abstracts the backend queue and uses a parametric wireless model; end-to-end wall-clock (with control-plane signaling for grid/codec metadata, CSR indices, and potential RANS entropy coding) is not dissected across diverse networks and straggler patterns. More real-network and asynchronous/many-client studies would strengthen the scaling claims.

- Comparisons against LLM baselines could be deeper.

**Questions:**

- Can you proofread the paper before the submission pls? lots of typos here,i.e., line 336 '?' and 'multi device' in the conclusion

- What is the control-plane overhead (bytes and latency) for metadata?

---

> ### Author Response · Authors · 2025-11-26
>
> # [Response to TyTH. Weakness 1]
>
> Thank you very much for the practical comment.
>
> - The concrete edge device used in the main experimental validation of our paper is the NVIDIA Jetson AGX Orin. This has been clearly mentioned in the revised version. Meanwhile, our framework (SLICER) is not specific to a certain type of edge device, but is applicable to any types of edge devices.
>
> - We also agree that in practice, the latency depends on the computation ability or specification of the edge device. To address this concern and ensure fairness in evaluation, in the revised version, we have expanded the experiments to cover a wide range of edge devices with diverse computational capabilities, including the Jetson TX1, Jetson Xavier NX, Jetson Orin NX, Jetson AGX Orin, RTX 2060, and RTX 3060. The results have been reported in Figure 7 of Appendix F in terms of both the SLICER's encoding/decoding latency and energy costs across diverse edge devices.
> While these metrics naturally scale with each device's computational capability, our framework maintains consistent efficiency, ensuring that both time and energy overheads remain within a practical and reasonable range across all test cases.
>
> ---
>
> # [Response to TyTH. Weakness 2 / Question 2]
>
> Thank you very much for the important comment and valuable suggestion. In the revised version, we have included analysis on the end-to-end wall-clock dissection across diverse networks and straggler patterns, along with real-network and asynchronous/many-client studies.
>
> **1. Control-Plane Signaling for Metadata, CSR, and rANS**
>
> In the revised version, we have analyzed the control-plane signaling overhead in the `Analysis of Metadata Overhead' section within Appendix G.2 and reported the results in Table 18.
>
> - **CSR Indices (84.0\% of packet):** The majority of metadata (235,200 bytes) consists of CSR column indices (\texttt{int}) required to map non-zero elements. This is an unavoidable cost of sparse representation.
>
> - **rANS Overhead (0.4\%):** The rANS entropy coding requires a frequency table for decoding. Since our quantization alphabet is small ($2^Q$), this header is extremely compact (approx. 1 KB).
>
> - Through the analysis, it turns out that metadata occupies 84.6\% of the compressed packet. This is because the payload itself is compressed just to 2.7\% of its original size via rANS. The critical metric is thus the absolute transmission volume, which is reduced by 82.2\% via our framework (SLICER) compared to the baseline.
>
> $$
> \\begin{array}{l|r|c|l}
> \\hline
> \\textbf{Component} & \\textbf{Size (B)} & \\textbf{Ratio} & \\textbf{Description} \\\\
> \\hline
> \\text{Control Plane} & 237,016 & 84.6\\% & \\text{Total Metadata} \\\\
> \\text{\\quad - CSR Indices} & 235,200 & 84.0\\% & \\text{Loc. of non-zero (int)} \\\\
> \\text{\\quad - CSR Rows} & 680 & 0.2\\% & \\text{Row pointers} \\\\
> \\text{\\quad - rANS Header} & 1,024 & 0.4\\% & \\text{Freq table} \\\\
> \\text{\\quad - Quant Params} & 72 & 0.03\\% & \\text{6 parts } \\times \\text{ (scale, zp)} \\\\
> \\text{\\quad - Shape Meta} & 40 & 0.01\\% & \\text{Tensor shape info} \\\\
> \\hline
> \\text{Payload (Val)} & 43,076 & 15.4\\% & \\text{Entropy-coded values} \\\\
> \\hline
> \\textbf{Total Size} & \\textbf{280,092} & \\textbf{100\\%} & \\textbf{5.6}\\times \\textbf{ Smaller than FP32} \\\\
> \\hline
> \\end{array}
> $$
>
> **2. End-to-End Wall-Clock Dissection**
>
> - In the revised manuscript, we have dissected and reported the full end-to-end wall-clock encoding/decoding latency of SLICER in Figure 7 (Appendix F), explicitly including the control-plane signaling with metadata, CSR, and rANS.
>
> - As can be seen from Figure 7, the total compute-side overhead of Jetson AGX Orin (i.e., the concrete edge device we used) remains below 10 ms for a reasonable range of IF sizes, which is thus negligible or insignificant compared to the transmission-latency reduction on bandwidth-limited links.

---

> ### Author Response · Authors · 2025-11-26
>
> # [Response to TyTH. Weakness 2 / Question 2] (Cont'd)
>
> **3. Studies on Diverse Networks and Straggler Patterns**
>
> - **Study on Diverse Networks:** In the revised version, we have included a study on diverse network conditions across different bandwidths. The results have been provided in Table 15 of Appendix F and demonstrate that SLICER-Search in our framework dynamically optimizes IF compression and adaptively adjusts the split point to satisfy latency constraints even under highly limited bandwidths (e.g., 2 MHz). This mechanism proves its stability in adverse network conditions without significant performance degradation.
>
> - **Study on Straggler Patterns:** By targeting a delivery probability of $1-\varepsilon$ (which accounts for ARQ retransmissions), the model inherently absorbs tail-latency spikes caused by channel noise and stragglers, as studied in [1].
>
> $$
> \\begin{array}{c|ccc|c}
> \\hline
> \\textbf{Sparsity} & \\textbf{10 MHz} & \\textbf{5 MHz} & \\textbf{2 MHz} & \\textbf{Acc.} \\\\
> & \\text{(ms)} & \\text{(ms)} & \\text{(ms)} & \\\\
> \\hline
> 0\\% & 724 & 1098 & 2055 & 74\\% \\\\
> 5\\% & 689 & 1045 & 1956 & 74\\% \\\\
> 15\\% & 621 & 942 & 1764 & 74\\% \\\\
> 25\\% & 554 & 840 & 1572 & 74\\% \\\\
> 35\\% & 484 & 734 & 1375 & 74\\% \\\\
> 45\\% & 415 & 630 & 1180 & 74\\% \\\\
> 55\\% & 346 & 525 & 983 & 73\\% \\\\
> 65\\% & 277 & 420 & 785 & 72\\% \\\\
> 75\\% & 206 & 312 & 584 & 71\\% \\\\
> \\hline
> \\end{array}
> $$
>
> [1] "Cooperative inference of DNNs for delay-and memory-constrained wireless IoT systems." IEEE Internet of Things Journal (2022).
>
> **4. Asynchronous/Many-Client Study**
>
> - **Many-Client Study:** In the revised version, we have expanded the evaluation to a testbed comprising multiple (and diverse) edge devices, RTX 3060 (6GB, INT4), RTX 2060 Super (8GB, INT8), RTX 3080 (12GB, INT8), and Jetson AGX Orin (32GB, FP16). The results have been reported in in Table 13 and show that utilizing edge devices with superior computational capabilities (e.g., RTX 3080 and Jetson AGX Orin) significantly improves the server efficiency. Specifically, in Scenario 3 (high-performance edge configuration), the server token load is reduced by 84.0\% and a 3.75$\times$ throughput speedup is achieved compared to the baseline, confirming that our framework effectively leverages the edge resources to alleviate the server workload.
>
> | Scenario | Latency (s) $\downarrow$ | Throughput $\uparrow$ | Server Tokens $\downarrow$ | Reduction (%) $\uparrow$ | Speedup $\uparrow$ |
> | :--- | :---: | :---: | :---: | :---: | :---: |
> | Baseline | 364.21 | 39.5 | 1,076,161 | - | 1.00$\times$ |
> | Scenario 1 | 320.41 | 44.9 | 928,291 | 13.7% | 1.14$\times$ |
> | Scenario 2 | 227.58 | 63.3 | 604,173 | 43.9% | 1.60$\times$ |
> | **Scenario 3** | **97.08** | **148.3** | **172,400** | **84.0%** | **3.75$\times$** |
>
> *Scenario Configs: **S1** (4$\times$3060), **S2** (1$\times$3060, 1$\times$2060S, 1$\times$3080, 1$\times$Orin), **S3** (2$\times$3080, 2$\times$Orin)*
>
> - **Asynchronous Study:** In our framework, no synchronization mechanism is required between front-end devices, which is indeed a significant merit in practice. Specifically, it is due to:
>
> - **Independent \& Stateless Operation:** Each device optimizes compression locally using SLICER-Search and requests asynchronously. The back-end treats incoming IFs as independent jobs (via standard queues or batching) without maintaining shared state or coordination across clients.
>
> - **Decoupled Control Plane:** Also, the updates to the latency target $D_i$ (to reflect server load) occur on a long-term (slow) timescale and do not require lock-step synchronization.
>
> In the revised version, we have explicitly stated in Sections 2.1 and 4.2 that front-end devices operate asynchronously and require no device-to-device coordination.
>
> **5. Real-Network Study**
>
> - The parametric wireless model ($\varepsilon$-outage model) adopted in our work is a realistic model [1]-[3] (and references therein) that accurately estimates the communication latency in practical wireless networks with (Gaussian) noises/inferences and channel fading (path loss).
>
> - Despite the practicality of the adopted wireless model, we agree that further study on real-network would strengthen our claims. Unfortunately, however, we believe that such study is beyond the scope of our current work due to the strict time limit of the rebuttal period.
> It should be an important next research step from our current work and deserved to be investigated in future works.
>
> [2] "DeCo-MeSC: Deep Compression-Based Memory-Constrained Split Computing Framework for Cooperative Inference of Neural Network." IEEE Transactions on Vehicular Technology (2025).
>
> [3] "Wireless channel adaptive DNN split inference for resource-constrained edge devices." IEEE Communications Letters (2023).

---

> > ### Author Response · Authors · 2025-11-26
> >
> > # [Response to TyTH. Weakness 3]
> >
> > Thank you very much for the comment. In the revised version, we have included in-depth comparative study by conducting additional experiments with diverse LLM baselines.
> > The results have been reported in Table 17 of Appendix F.
> >
> > - The results demonstrates the SLICER's consistent and stable performance across various LLM architectures and sizes, even under aggressive compression (Sparsity $s=0.7$, Q=[8,8,8]).
> >
> > - Notably, for certain LLM baselines such as Qwen-Chat-0.5 (+3.66\%) and Qwen-Chat-7B (+0.63\%), performance gains (rather than degradation) are observed. This observation aligns with the trends seen from the results across diverse tasks and architectures in Table 3.
> >
> > **Table: Performance comparison on HumanEval and GSM8K benchmarks.**
> > *(Comparison with Baseline, $\Delta$ indicates accuracy change)*
> >
> > | Model | Benchmark | Acc. (%) | $\Delta$ (pp) |
> > | :--- | :--- | :---: | :---: |
> > | **Phi-3-mini** | HumanEval | 63.41 | $\color{blue}{-0.61}$ |
> > | **Phi-3-med** | HumanEval | 50.00 | 0.00 |
> > | **Gemma-2b** | HumanEval | 28.05 | $\color{blue}{-1.83}$ |
> > | **Llama-3.2-1B** | HumanEval | 32.32 | $\color{blue}{-3.66}$ |
> > | **Qwen-0.5** | HumanEval | 34.15 | $\color{red}{+1.22}$ |
> > | **Qwen-Chat-0.5** | HumanEval | 51.22 | $\color{red}{+3.66}$ |
> > | **Qwen-Chat-7B** | HumanEval | 82.32 | $\color{red}{+0.63}$ |
> > | **Llama-3-8B** | HumanEval | 54.88 | $\color{red}{+1.83}$ |
> > | **DeepSeek-Math** | GSM8K | 81.05 | $\color{blue}{-0.75}$ |
> > | **Llama-3-8B** | GSM8K | 64.14 | $\color{blue}{-0.61}$ |
> >
> > - It manifests that our framework is task-agnostic and architecture-agnostic across diverse LLM backbones and benchmarks while maintaining (in some cases, improving) generation quality with significant compression.
> >
> > ---
> >
> > # [Response to TyTH. Question 1]
> >
> >
> > We thank Reviewer for pointing this out and for their careful reading. We apologize for the typographical errors. In the revised version, suggested fixes have been addressed accordingly. Also, we have performed a thorough proofread of the entire manuscript to correct all identified typos, grammatical errors, and presentation issues to improve the paper's clarity.

---

> ### Comment · Reviewer_TyTH · 2025-11-28
>
> I highly appreciate the additional experiments added by the author and the efforts the authors making on the rebuttal. As such, I will increase my final score once the system bugs are fixed.

---

> > ### Author Response · Authors · 2025-11-28
> >
> > Thank you for your kind words and for taking the time to reconsider your evaluation. We're glad our responses helped clarify the concerns, and we truly appreciate your updated assessment.

---

### Meta-Review · Area_Chair_o5ic · 2026-01-10

**Summary:**

The three reviewers have raised a range of issues for the submission, and the authors have actively engaged with the reviewers by providing detailed answers, which resulted in one reviewer (TyTH) to raise the rating.

But it seemed that all reviewers were not very interested in raising the rating further even after the discussion, resulting a flat rating of 4 for all three reviwers.

Given the final rating from three reviewers and the consistency among their rating, I decided to keep the same rating and recommend a reject for this submission.

**Reviewer Concerns:**

The reviewers raised a range of issues for the work, such as:

•	The evaluation is not accurate enough without detailed dissection of delays across different stacks
•	Comparison with prior work is lacking [BootleFit and Frankensplit]
•	Lack of diversity in device evaluation
•	The need to explore multi-point portioning beyond single-point splitting
•	The need to analyze multi-client scenarios
•	Lack of analysis of the long token setting
•	Lack of profiling of device-side computation/memory overhead and the scalability concerns
•	Lack of rigorous in choosing certain variables

The authors have also provided some explanations for all the issues raised. But there seemed to be a lack of inclinations from reviewers to further increase their rating.

**Reviewer Scores:**

One reviewer (TyTH) has raised the rating, but it seemed to be at 4. The other reviewers may not change their rating.

---

### Decision · Program_Chairs · 2026-01-26

Reject